# Changes in the Heart Rate of Sniffer Dogs Trained for Detection of Lung Cancer

**DOI:** 10.3390/diagnostics13152567

**Published:** 2023-08-01

**Authors:** Petra Riedlova, Spiros Tavandzis, Josef Kana, Silvie Ostrizkova, Dagmar Kramna, Libor Krajcir, Tereza Kanova, Simona Lastikova, Hana Tomaskova, Jaromir Roubec

**Affiliations:** 1Department of Epidemiology and Public Health, Faculty of Medicine, University of Ostrava, 703 00 Ostrava, Czech Republic; silvie.ostrizkova@osu.cz (S.O.); dagmar.kramna@osu.cz (D.K.); hana.tomaskova@osu.cz (H.T.); 2Centre of Epidemiological Research, Faculty of Medicine, University of Ostrava, 703 00 Ostrava, Czech Republic; 3Czech Centre for Signal Animals, 741 01 Novy Jicin, Czech Republic; 4Laboratory of Molecular Biology, Department of Medical Genetics, AGEL Laboratories, 741 01 Novy Jicin, Czech Republic; 5Department of Pneumology and Phthisiology, University Hospital with Polyclinic FDR Banska Bystrica, 975 17 Banska Bystrica, Slovakia; 6Department of Pulmonary, Vitkovice Hospital, 703 00 Ostrava, Czech Republic; jaromir.roubec@vtn.agel.cz

**Keywords:** lung cancer, sniffer dog, detection, diagnosis, heart rate

## Abstract

Background: Lung carcinoma is one of the most common malignancies worldwide. At present, unfortunately, there are no markers that would allow early identification of this tumor in the preclinical or early clinical stage. The use of sniffer dogs has been reported to show some promise in early diagnosis of this type of cancer Aim: This study aimed to evaluate the possibility of utilizing changes in the heart rate of sniffer dogs (which increases when finding a positive sample) in tumor detection. Methods: This double-blinded pilot study included two sniffer dogs. A chest strap was fastened on the dog’s chests for heart rate monitoring while they were examining samples and heart rate was recorded. Test parameters (sensitivity, specificity, positive and negative predictive values) were then calculated, evaluating performances based on (i) the dog’s indications according to their training and (ii) the changes in their heart rates. Results: Calculation according to the dog’s indications revealed an overall sensitivity of 95.2% accompanied by a specificity of 81.8%, a PPV of 93.7%, and an NPV of 85.7%, respectively. These results were not significantly different from those evaluated by heart rate; heart rate monitoring was, however, burdened with a relatively high proportion of invalid experiments in which heart rate measurement failed. When the method of calculation was changed from rounds to individual samples, the test parameters further increased. Conclusions: This pilot study confirmed the hypothesis that heart rate increases in trained sniffer dogs when encountering samples from tumor-positive patients but remains unchanged when only negative samples are present. The reliability of results based on heart rate increase is similar to that obtained by a dog’s indications and, if the limitation represented by technical issues is overcome, it could serve as a valuable verification method.

## 1. Introduction

Year by year, lung carcinoma is one of the most common malignancies worldwide. In 2020, approximately 2,206,771 new cases were identified. In Western countries, we can observe a decreasing trend due to the gradually decreasing number of smokers. In developing countries, however, the opposite can be seen [1,2]. In 2018, lung carcinoma was globally the cause of 1,761,000 deaths, which represents 18.4% of cancer deaths worldwide [1,3].

At present, unfortunately, there are no markers that would allow the early identification of this tumor in the preclinical or early clinical stage. Diagnosis relies predominantly on imaging methods, such as lung X-ray, computed tomography (CT), or a more efficient low-dose helical computed tomography (LDCT) [4]; the latter, however, is prone to false positive results as it is known to also detect non-malignant abnormalities [5]. Sputum cytology is a useful tool for the detection of tumors in the major bronchi; it is, however, unsuitable for the detection of smaller adenocarcinomas (<2 cm in diameter) in minor bronchi, bronchioles, and alveoli. Bronchoscopy with lung biopsy and histological evaluation follows to confirm the diagnosis [6]. Still, a majority of tumors are diagnosed only in an advanced stage, which leads to a 5-year survival of only 11.2% in men and 13.9% in women [7]. Timely detection of tumors is, therefore, of utmost importance.

Early diagnosis of selected tumors is the subject of research by the Czech Center for Signal Animals, specializing in the training of signal dogs. These dogs are trained for early detection of selected tumor types from the patient’s blood samples. The Center deals with tumours that are difficult to diagnose with medical instruments in the early stages (e.g., lung cancer and ovarian cancer [8]. The principle lies in the dog’s smell being several orders of magnitude more sensitive than diagnostic instruments [9]; thanks to this, dogs are capable of identifying volatile organic compounds produced by tumor metabolism [10]. It is not yet known what specific compounds are contained in the tumour. But it is four–five thousand different molecules [8]. The dogs are trained to identify the samples from people with cancer by performing a certain activity (e.g., the dog sits down or lies down in front of the respective sample). This method is, however, considered by some as insufficiently proven and insufficiently objectivized [11].

Hence, in this study, we hypothesized that objectivization of the process through monitoring heart rate (HR) changes of trained sniffer dogs during sniffing experiments could improve the performance. The hypothesis was based on the assumption that if the dog “suspects” a sample but is not sure enough to tag the sample as positive, its HR would go up, as it does when finding a clearly positive sample and expecting a reward. The same phenomenon was observed in avalanche dogs whose heart rate increased when detecting a human being [12].

We expected that the detection based on HR increase could theoretically lead to an increased false positive rate; however, as this type of error is less serious than the opposite (failure to detect a positive sample) and would only lead to a more thorough examination rather than to omitting an already existing tumor, it appears that the expected tradeoff of improved sensitivity at the expense of slightly reduced specificity would be acceptable. The hypotheses of this research were therefore as follows: (i) trained dogs can effectively recognize the difference between the negative and positive samples, (ii) monitoring dogs’ HR during action can be a useful auxiliary method for improving the sensitivity and negative predictive value compared to the standard performance when the dog just indicates samples from cancer patients. This study is the first one attempting to objectivize the detection process using heart rate.

## 2. Methods

This pilot study included two sniffer dogs (an Australian cattle dog and a German Shepherd) selected from the pool of dogs trained in the Czech Center for Signal Animals as the two individuals that best tolerated wearing the HR monitor on their bodies. The dogs were trained using intermittent reinforcement as described in detail in our previous publication [8]. In addition to the standard conditions used for training, a chest strap was fastened around the chests of the dogs, which allowed continuous HR monitoring using the smartwatch SUUNTO Ambit3 (Vantaa, Finland) and the supplier-produced software (version 2.26.1).

The study was designed as a double-blinded study, with neither the trainer nor the owner of the dog knowing the status of individual samples. Blood samples from individuals ≥18 years of age with histologically confirmed lung carcinoma (regardless of the stage, sex, or age) were considered positive, and negative samples were acquired from individuals without a confirmed diagnosis of lung cancer. No other inclusion/exclusion criteria were applied. All participants whose blood samples were used for training or experiments signed informed consent approved by the local Ethics Committee.

The samples were prepared using blood serum from 5 mL of full blood taken from the participants. The blood was centrifuged at 4000 RCF/4 °C for 10 min and the resulting supernatant (serum) was pipetted into a storage vial and stored in a freezer at −12 °C. Before use, the serum was thawed and, after shaking to thoroughly mix the contents of the storage vial, one drop was placed on the bottom of another vial. A pad of cotton wool was inserted into the new vial above the serum (but not in direct contact) as an odor adsorbent and the vial was kept closed for 24 h [8]. After that, the “scented” pads were placed separately into closed vials without blood serum and stored until experiments.

The experiments themselves were performed as follows: for each round (i.e., each release of the dog for detection), four vials with samples were placed into a stainless steel tray with 4 holes laying on the ground. For each individual round (i.e., an individual sample presentation to a dog), a set of 4 samples was prepared by the administrator prior to the experiments (3 negative samples and one “unknown” sample; that sample was taken either from the pool of negative samples, or from the pool of positive samples). During experiments, the dog owner/trainer placed samples into the four holes in the tray (as the owner/trainer was blinded to the content of the vials, they were placed at random positions within the tray). Subsequently, the dog was released to examine the vials with pads. Each round was video recorded and the initial and highest heart rates during the individual presentation (which took approx. 10–20 s in each case) were written down, as well as the result of the dog’s indication (or not) of the sample positivity. None of the persons present during the experiments were aware of the positivity/negativity of the samples. The two dogs alternated in their rounds after 10 min (i.e., approximately after 10 samples, including sample placement) to allow sufficient time for regeneration and rest. None of the samples used in the training were used for experiments; in addition, none of the samples used during the experiments were presented twice to the same dog.

After the experimental part of the study was completed, the results were unblinded and evaluated based on (a) dogs’ indication and (b) heart rate increase during the experiment. Any HR increase above the initial value (base HR) was considered an indication of the presence of the positive sample in the set. Although the HR kept gradually changing over the course of the experiments, preliminary testing showed us that dogs’ heart rates increased only when the dogs were excited for any reason. From this perspective, changes in the actual base heart rate at the beginning of the individual round played no role in the evaluation as only the increase in HR during the round was considered. The experimental setting was designed in such a way that there was no potential source of excitement other than the samples. Basic test parameters were subsequently calculated, namely, sensitivity (SEN), specificity (SPE), and positive (PPV) and negative (NPV) predictive values. Both of these classes were, in addition, evaluated in two ways: (i) considering each experiment separately (i.e., each set of 4 samples) as one round with a positive/negative result, and (ii) considering each sample separately (i.e., considering each round as four independent samples; for example, an experiment with 4 negative samples where the dog indicated none of the samples was considered as 4 correctly identified negative samples). The reasons for this approach are discussed below in the Section 4.

## 3. Results

### 3.1. Overall Performance and Test Parameters

The demographic parameters of the donors of positive samples are shown in Table 1.

Out of 115 experiments performed during the study, 84 were valid (i.e., heart rate monitoring worked without problems) and were used for analysis. In 62 experiments, a positive sample was present in the sample set, while in 22 cases, only negative samples were present. The dogs correctly indicated positivity in 59 (out of 62) positive “unknown” samples and correctly indicated no sample in 18 (out of 22) sets with negative samples. Of the three failures where positive samples were present but not recognized, an incorrect sample (a negative one) was indicated by the dog as positive in two cases, and in one case, no sample was tagged by the dog. Among the negative samples, four false positives were recorded (see Table 2).

Table 2 also contains the results of the HR-based evaluation. The number of true positives was somewhat lower; interestingly, on three occasions, the dog’s HR did not increase even though the dogs indicated the sample correctly. On the other hand, in one round, the dog’s HR increased, even though it failed to indicate a positive sample.

Furthermore, an additional calculation of test parameters considering each sample separately (rather than each round) was performed (see Table 3). Note that this calculation was only possible for indications by the dog, as the change in the heart rate did not identify a particular sample but only the presence of a positive sample in the round.

The resulting test parameters are shown in Table 4. The sensitivity of 95.2% calculated according to the dog indications was accompanied by a specificity of 81.8%, a PPV of 93.7%, and a NPV of 85.7%. These results were generally slightly, though insignificantly (*p* = 0.763), better than those evaluated by heart rate only (sensitivity of 91.9%, specificity of 81.8%, a PPV of 93.4%, and a NPV of 78.3%).

When the method of calculation was changed from rounds to individual samples, the test parameters further increased (which is logical as the number of negative samples grew significantly; see Table 3 and Table 4).

The results of both dogs participating in the experiments were also compared (see Table 5), showing the insignificantly better performance of the Australian cattle dog.

### 3.2. Heart Rate Changes during Individual Experiments

The baseline HR was changing over time; in addition, the baseline HR slightly differed between dogs as well (Table 6). The observed results (a typical record is shown in Figure 1), however, confirmed that any increase in HR during a round was consistent with the identification of positive samples; the heart rates often tended to decrease during the rounds with negative samples (or incorrectly identified positive samples).

Figure 2 compares the dogs’ initial heart rates and the maximum heart rates in the rounds, presenting heart rates separately for positive and negative samples. It is obvious that while the HR did not change significantly (*p* = 0.94) during rounds containing only negative samples, a significant change (*p* < 0.001) was observed when a positive sample was present in the sample set.

## 4. Discussion

The presented study aimed to investigate the success of the trained sniffer dogs in detecting lung carcinoma and to compare the results of dogs’ indications to those obtained by heart rate measurements.

Although heart rate measurement counts among basic physical examinations, information about the normal heart rate of dogs is scarce. The high physiological and psychological variability of dog breeds makes the use of an arbitrary HR impossible [13], with physiological values generally ranging between 70–120 beats per minute (bpm) [14]. Moreover, although HR correlates with body weight in many mammalians, this does not consistently apply to dogs [15,16].

Recently, the use of human heart rate monitors for the measurement of dogs’ heart rates [12,17,18,19] has become preferred to the gold standard, i.e., electrocardiography (ECG) [20,21]. Such monitors consist of a chest strap with electrodes and a built-in signal transmitter. The application of a conductive gel on the electrodes before fastening the chest strap on the dog helps improve the reliability of the function of the belt. The data from the strap are forwarded to a smartwatch and can be analyzed on a computer. The advantages of such monitors compared to ECG include dog-friendliness (no fur shaving is necessary), easy mobility, simple application, and lower costs. Studies comparing these two methods revealed a good agreement of the results of HR monitors with ECG in dogs [12,18,19,22].

However, the problematic functionality of the HR monitor in some cases was a downside of the use of the HR monitor in our study. In all, 115 experiments with HR measurement were performed, of which HR was properly recorded only in 84 experiments. This was mostly associated with the fact that the HR monitors are constructed to fit and work with human skin, and the fur interfered with their function in 33% and 20% of the measurements with individual dogs, when, despite the application of the conductive gel, the monitors lost contact with the skin (Table 6). In some studies on monitoring dogs’ heart rates for other purposes, such as medical ones, researchers shaved the hairs from the dog’s chests to improve the contact between the electrodes and the skin [12,18]; in our research, however, we opted for a more dog-friendly approach as we were still able to acquire a sufficient number of valid measurements, and shaving could possibly stress the dogs, interfering with their performance. Moreover, Bidoli et al. reported that the fur itself does not influence the effectiveness of the chest strap as, curiously, a higher frequency of invalid measurements was found in short-haired than long-haired dogs [19].

In addition to the fur, we hypothesize that the dog’s size is another important factor—the percentage of valid experiments was higher in the larger German Shepherd than in the somewhat smaller Australian cattle dog (80%, resp. 67%); this could be caused by differences in the fit of the strap constructed for the much larger human chest. This is, however, in contrast with results by Bidoli et al., who stated that higher chest circumference was associated with a higher frequency of invalid results [19]. Some researchers also use gauze for additional fastening of the chest strap to the dog’s chest, which can further improve performance [12,18]. We have, however, rejected this option as well in order to prevent the potential stress that could affect the results.

In our study, 62 valid rounds with positive samples (i.e., three negative and one positive sample in the round) and 22 rounds with solely negative samples (four per round) were performed. This setup led to the underestimation of specificity and negative predictive values (note that in the used setting, where only one sample is unknown and all others are negative, three samples in each round are “disregarded”; Table 2). For this reason, we have performed an additional calculation considering each sample separately (Table 3). This allowed a finer analysis and correction of some issues that could not be addressed during the “by rounds” calculation. For example, if the dog indicates an incorrect sample in a round containing a positive sample, it is considered a false-negative when calculated per rounds, but in the finer calculation based on individual samples, it can be considered a false negative (the positive sample was not identified), a false positive (a negative sample was indicated as positive), and two true negatives (two negative samples that were correctly ignored). This calculation, however, could not be used in the case of heart rate as it is not possible to determine which sample causes the heart rate to increase (Table 4). This duality of calculations can be considered both a limitation of the study and its strength as it facilitates a complex evaluation from both perspectives. To eliminate this duality, it would be necessary to evaluate only one sample per round instead of a set of four; this, however, would require a completely different strategy of training (a new approach from the beginning of training; dogs already trained for runs of four samples cannot be retrained this way) and might be difficult to implement as it is assumed that the dogs need to have a negative sample in the set of tested samples to be better able to recognize the positive sample. Moreover, the approach with multiple samples is a standard setting commonly used in studies with sniffer dogs [23,24].

We should also mention that some false positive samples were marked as false positives repeatedly by both dogs. Such samples are being recorded and the patients will be subject to more thorough follow-ups as it is possible that the dogs have detected forming tumors at a very early stage preceding the clinical diagnosis.

There is another peculiarity that needs to be mentioned; when evaluated according to the dog’s indications and rounds, the dogs indicated another sample in the set on two out of three occasions in which they failed to recognize a positive sample. This may suggest that the dogs were aware of the presence of a positive sample in the set but failed to identify the correct one. This was also associated with another observation—from our experience (including training), we know that dogs are more likely to err in freshly scented samples than in older ones; all positive samples that the dogs failed to recognize were less than 2 weeks old. This may also have implications for practice and letting the samples “mature”, or repeating the identification with a “matured” sample after several weeks might further improve the results.

In this type of screening, sensitivity and negative predictive value are probably the most important parameters (false positives playing a key role in calculations of specificity and PPV are not as problematic, as they would only lead to a more thorough examination and follow-up of patients, which is far less serious than false negativity, i.e., failure to identify a patient with cancer). From this perspective, a sensitivity of 95% based on the dogs’ indications is a very good result, slightly better (though not significantly, *p* = 0.763) than results based on the heart rate. Negative predictive values of 85.7% (dog’s indications) and 78.3% (heart rate) in the calculations per rounds are not so favorable; we must, however, take into account the aforementioned fact that in every run, there were three or four negative samples, so calculations per rounds underestimate the actual NPV. Once individual samples were considered, the NPV grew to an excellent value of almost 99%. Of course, we have to take into account that NPV is greatly affected by prevalence, and that by considering all samples individually, we have altered that prevalence. Still, this high number accompanied by the sensitivity of 95% indicates a good potential of this method for future use in clinical practice.

The confidence intervals of some parameters (specificity and NPV in evaluation by rounds) are relatively wide. This is caused by the relatively low number of experiments with false negative/positive results in this experimental setup.

Our results of evaluation based on dogs’ indications are similar compared to other studies. Examples include studies by Ehmann et al., McCulloch et al., Elliker et al., Cornu et al., Kitiyakara et al., and Guerrero Flores et al. (see Table 7; [25,26,27,28,29,30]).

The hypothesis that the HR would increase if the dog finds a positive sample was confirmed (Figure 2). However, the assumption that it would increase even when the dog is uncertain about the sample and, hence, the overall sensitivity would improve, was not proven true. In effect, as the use of HR instead of the indication by dogs did not improve the results, and considering the complications associated with the chest strap resulting in a high frequency of invalid experiments, we cannot recommend the use of HR monitoring as a parameter superior to the indications by trained sniffer dogs.

## 5. Conclusions

The results of the presented pilot study indicate that the heart rates of trained sniffer dogs significantly increase when the dog finds a sample originating from a person with cancer, while when only negative samples are present in the set, the HR remains stable. Nevertheless, the indication by the trained dogs was not inferior to the HR-based evaluation. In view of that, the fact that HR measurements are associated with technical problems resulting in a relatively high number of invalid experiments, and in view of the simplicity of the use of indications by the trained sniffer dogs appears to be an overall better choice. However, if the way of the training is modified so that the dog can work with a single sample per round, HR monitoring could become an important supplementary method. There are also other possibilities that could support this detection, such as the overall behaviour of the dog during exercise. However, further experimental planning and observations are required. The results of this study can contribute towards the improvement of the early diagnosis and, thus, the outcome of patients with lung cancer.

## Figures and Tables

**Figure 1 diagnostics-13-02567-f001:**
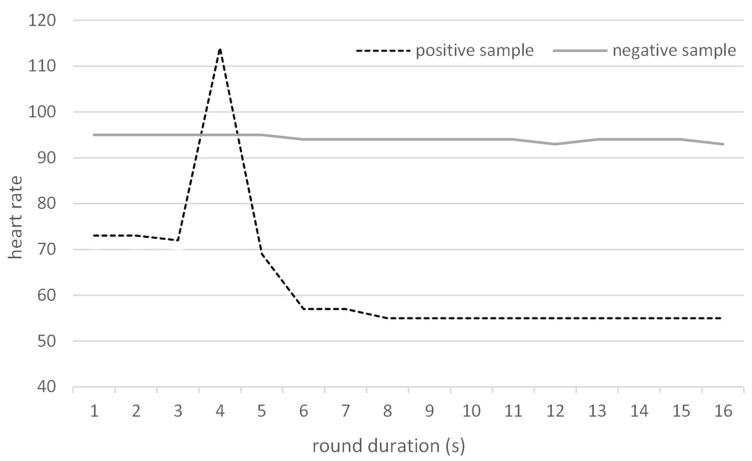
A typical continuous dog’s HR record during experiments including a positive sample or only negative samples.

**Figure 2 diagnostics-13-02567-f002:**
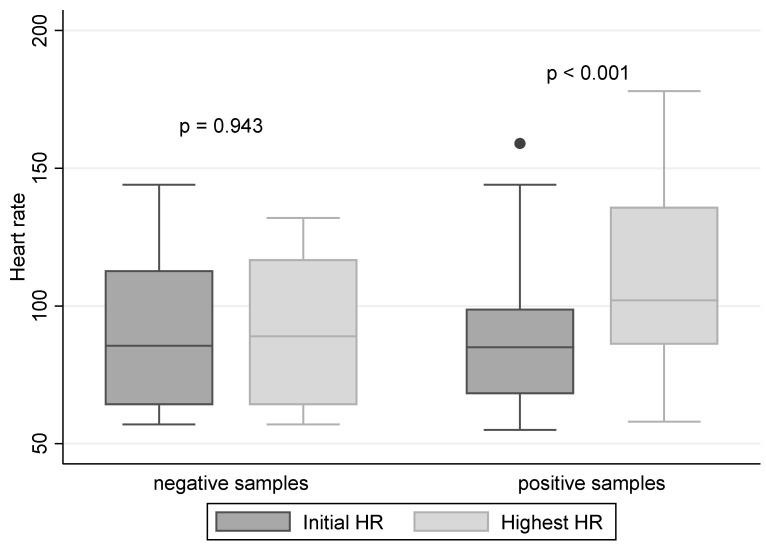
Changes in HR during experiments; the horizontal line indicates the median HR, the box indicates the interquartile range, and the individual points indicate the outliers; *p*-value is the results of the Wilcoxon signed-rank test.

**Table 1 diagnostics-13-02567-t001:** The demographic parameters of the donors of positive samples.

Lung Tumor Type	Adenocarcinoma	Squamous Cell Carcinoma	Small Cell Carcinoma (SCLC)
Number of patients	16	24	20
Stage 1	2	0	0
Stage 2	5	4	3
Stage 3	2	10	12
Stage 4	7	10	5
Total average age	64.14	69.70	67.60
Women	7	9	13
Men	9	15	7

**Table 2 diagnostics-13-02567-t002:** Results evaluated according to the indication by the trained dogs and by their HR, with entire rounds (4 samples each) considered.

Evaluation According to the Indication by the Dogs (Evaluated by Rounds)			
Rounds with a positive sample (+3 negatives)	62	True positives (test positives condition present)	59
		False negatives (test negatives condition present)	3
Rounds with negative samples only	22	True negatives (test negatives condition absent)	18
		False positives (test positives condition absent)	4
Evaluation according to the heart rate (evaluated by rounds)			
Rounds with a positive sample (+3 negatives)	62	True positives	57
		False negatives	5
Rounds with negative samples only	22	True negatives	18
		False positives	4

**Table 3 diagnostics-13-02567-t003:** Results of indications by dogs—individual samples considered.

Positive Samples		Negative Samples	
62		274	
True positives	False negatives	True negatives	False positives
59	3	268	6

**Table 4 diagnostics-13-02567-t004:** Test parameters calculated based on the indications by dogs and by heart rate.

		Dog Indications-Based Calculations (95% CI)	Heart Rate-Based Calculations	*p*
Calculation by rounds	Sensitivity	95.2% (85.6–98.7)	91.9% (81.5–97.0)	0.717
	Specificity	81.8% (59.0–94.0)	81.8% (59.0–94.0)	1.000
	PPV	93.7% (83.8–98.0)	93.4% (83.3–97.9)	1.000
	NPV	85.7% (62.6–96.2)	78.3% (55.8–91.7)	0.701
Calculation by samples	Sensitivity	95.2% (85.6–98.7)	Not applicable	
	Specificity	97.8% (95.0–99.1)		
	PPV	90.8% (80.3–96.2)		
	NPV	98.9% (96.5–99.7)		

PPV—Positive predictive value, NPV—Negative predictive value.

**Table 5 diagnostics-13-02567-t005:** Comparison of the results of both dogs used in the experiment.

		Australian Cattle Dog (95% CI)	German Shepherd (95% CI)	*p*
Calculation by rounds	Sensitivity	96.7% (81.0–99.9)	93.8% (77.8–98.9)	1.000
	Specificity	90.0% (54.1–99.5)	75.0% (42.8–93.3)	0.594
	PPV	96.7% (81.0–99.9)	90.9% (74.5–97.6)	0.614
	NPV	90.0% (54.1–99.5)	81.8% (47.8–96.8)	1.000

**Table 6 diagnostics-13-02567-t006:** Heart rates of both dogs.

	Australian Cattle Dog	German Shepperd
Rounds	40	44
% valid rounds	67%	80%
Minimum recorded heart rate	55	57
Maximum recorded heart rate	178	148

**Table 7 diagnostics-13-02567-t007:** Evaluation based on dogs’ indications in other studies.

Reference	Number of the Tested Cases	Sensitivity	Specificity	PPV	NPV
Ehmann et al. (2012) [25]	25	72%	94%	75%	93%
McCulloch et al. (2006) [26]	28	99%	99%	x	x
Elliker et al. (2014) [27]	50	75%	71%	x	x
Cornu et al. (2010) [28]	33	91%	91%	x	x
Guerrero-Flores et al. (2017) [30]	50	93%	99%	x	x

## Data Availability

The raw data supporting the conclusions of this article will be made available by the authors, without undue reservation.

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
