# Peer review of "Changes in the Heart Rate of Sniffer Dogs Trained for Detection of Lung Cancer"

_diagnostics, 2023, doi:10.3390/diagnostics13152567_

Round 1

Reviewer 1 Report

pilot experimental study ReidlovA et al. presented the results of HR in trained dogs for the early detection of lung cancer. I consider the study to be methodologically well-executed, and the results are clearly presented with highlighted limitations and an interesting discussion. I congratulate the authors, I wish them continued success in their research. Anyway, I have the following questions: 1. Are the healthy subjects completely healthy, and not just from lung cancer? That is, can the existence of some other tumor or disease give a false positive result? 2. How do you propose to solve the technical problems related to the placement of the electrodes and to what extent do the electrodes and the test itself affect HR in dogs?
Best wishes!

Author Response

Reviewer question 1

pilot experimental study ReidlovA et al. presented the results of HR in trained dogs for the early detection of lung cancer. I consider the study to be methodologically well-executed, and the results are clearly presented with highlighted limitations and an interesting discussion. I congratulate the authors, I wish them continued success in their research. Anyway, I have the following questions:

  1. Are the healthy subjects completely healthy, and not just from lung cancer? That is, can the existence of some other tumor or disease give a false positive result?

Author's response

Thank you very much for your questions. Individuals are completely healthy, or they have another type of disease that is not cancerous. But, of course, we cannot rule out the possibility that people have another type of cancer that they do not know about. However, we specialise dogs for a given type of cancer, so we assume (and test) that the dog will respond only to the type of cancer it is taught to respond to. But we cannot yet say with 100% certainty that this is the case. Many more experiments are still needed.

Reviewer question 1

  1. How do you propose to solve the technical problems related to the placement of the electrodes and to what extent do the electrodes and the test itself affect HR in dogs?

Author's response

We suggest that next time we wrap a bandage around the electrodes. We could also choose other breeds of dogs with a larger chest or long hair. These dogs were short-haired. Shaving the hair at the electrode insertion site is also a solution. We wanted to avoid this.

The electrodes themselves did not have an effect on our dogs. Even when they were resting, they had the electrodes attached and their HR did not change. If they were stressed about it, their HR would have changed even during rest. Before and after work, the dogs' HR was stable.

Reviewer 2 Report

This study was interesting about the detection of lung cancer by the dog's sense of smell.

However, I have a few questions.

1.     Why does the heart rate increase in positive samples? Is it because of the components present in positive samples that cause the increase in heart rate, which is difficult to consciously control in humans?

2.     Are there any other observable indicators, such as differences in barking patterns?

3.     In positive samples, what specific compounds does the dog's sense of smell detect? Do these compounds differ depending on the type or stage of cancer?

4.     Does olfactory fatigue not occur in the dogs during the detection process?

5.     How long does it take to make a detection?

6.     Can dogs detect any type of cancer?

Author Response

Reviewer question 2

This study was interesting about the detection of lung cancer by the dog's sense of smell.

However, I have a few questions.

  1. Why does the heart rate increase in positive samples? Is it because of the components present in positive samples that cause the increase in heart rate, which is difficult to consciously control in humans?

Author's response

Thank you very much for your questions.

This is probably because the dog is euphoric and happy when the correct positive sample is found. And he knows that he will receive a reward (either in the form of praise or a treat) for finding the right one. That is why we use unblinded samples that we know are positive or negative in the first phase of training, but also as maintenance training. This is to make sure that the dog is really looking for what he has.

Reviewer question 2

  1. Are there any other observable indicators, such as differences in barking patterns?

Author's response

I am sure there are. In one of our next experiments, we would like to look at the behaviour of the dog during training (expression etc.). But it is a lot of work to prepare and evaluate afterwards.

Reviewer question 2

  1. In positive samples, what specific compounds does the dog's sense of smell detect? Do these compounds differ depending on the type or stage of cancer?

Author's response

It is not yet known what specific compounds are contained in the tumour. But it is 4-5 thousand different molecules. If it were known, electronic noses would develop with zero error rate.  Depending on the type of cancer, the "smell" certainly varies, because when a dog is trained for one type of tumour, it does not care about the other. And as for the stage of the cancer, it is more likely to be about the different intensity of the smells of the different compounds.

Reviewer question 2

  1. Does olfactory fatigue not occur in the dogs during the detection process?

Author's response

Olfactory fatigue probably does not occur, but mental fatigue does. This is a demanding activity for the dog, so he must have regular breaks between training, e.g. after 10 minutes of training - 15 minutes break. But it is very individual for each dog.

Reviewer question 2

  1. How long does it take to make a detection?

Author's response

Detection by one dog alone can take from a few seconds to a few minutes, but this detection is done by multiple dogs to ensure that one dog does not have to have "his day". After all, it is not a machine, it is a living being.

Reviewer question 2

  1. Can dogs detect any type of cancer?

Author's response

They can only detect the type of tumour that they are trained to detect. However, we only treat tumours that are difficult to diagnose with medical instruments and are usually detected in late stages. such as lung cancer and ovarian cancer. In these cases, we see this prediagnosis as a perfect complementary method.

Round 2

Reviewer 2 Report

I understand the contents of the author's reply, but I do not know what part of the manuscript was revised.  It can be accepted if properly revised.

Author Response

Reviewer question 2

This study was interesting about the detection of lung cancer by the dog's sense of smell.

However, I have a few questions.

  1. Why does the heart rate increase in positive samples? Is it because of the components present in positive samples that cause the increase in heart rate, which is difficult to consciously control in humans?

Author's response

Thank you very much for your questions.

This is probably because the dog is euphoric and happy when the correct positive sample is found. And he knows that he will receive a reward (either in the form of praise or a treat) for finding the right one. That is why we use unblinded samples that we know are positive or negative in the first phase of training, but also as maintenance training. This is to make sure that the dog is really looking for what he has.

Described in the introduction (marked in red). “The hypothesis was based on the assumption that if the dog “suspects“ a sample but is not sure enough to tag the sample as positive, its HR would go up, just like when finding a clearly positive sample and expecting a reward. The same phenomenon was observed in avalanche dogs whose heart rate increased when detecting a human being.“

Reviewer question 2

  1. Are there any other observable indicators, such as differences in barking patterns?

Author's response

I am sure there are. In one of our next experiments, we would like to look at the behaviour of the dog during training (expression etc.). But it is a lot of work to prepare and evaluate afterwards.

Added to the Conclusion:(marked in red).  “However, if the way of the training is modified so that the dog can work with a single sample per round, HR monitoring could become an important supplementary method. There are also other possibilities that could support this detection, such as the overall behaviour of the dog during exercise. However, further experimental planning and observations are required. The results of this study can contribute towards the improvement of the early diagnosis and, thus, the outcome of patients with lung cancer.”

Reviewer question 2

  1. In positive samples, what specific compounds does the dog's sense of smell detect? Do these compounds differ depending on the type or stage of cancer?

Author's response

It is not yet known what specific compounds are contained in the tumour. But it is 4-5 thousand different molecules. If it were known, electronic noses would develop with zero error rate.  Depending on the type of cancer, the "smell" certainly varies, because when a dog is trained for one type of tumour, it does not care about the other. And as for the stage of the cancer, it is more likely to be about the different intensity of the smells of the different compounds.

Added to the Introduction: (marked in red).  “It is not yet known what specific compounds are contained in the tumour. But it is 4-5 thousand different molecules.“

Reviewer question 2

  1. Does olfactory fatigue not occur in the dogs during the detection process?

Author's response

Olfactory fatigue probably does not occur, but mental fatigue does. This is a demanding activity for the dog, so he must have regular breaks between training, e.g. after 10 minutes of training - 15 minutes break. But it is very individual for each dog.

Specified in the Methods: (marked in red). “The two dogs alternated in their rounds after 10 minutes (i.e., approximately after 10 samples, including sample placement) to allow sufficient time for regeneration and rest.“

Reviewer question 2

  1. How long does it take to make a detection?

Author's response

Detection by one dog alone can take from a few seconds to a few minutes, but this detection is done by multiple dogs to ensure that one dog does not have to have "his day". After all, it is not a machine, it is a living being.

Specified in the methods.(marked in red).  „Each round was video recorded and the initial and highest heart rates during the individual presentation (which took approx. 10-20 s in each case) were written down, as well as the result of the dog’s indication (or not) of the sample positivity.“

Reviewer question 2

  1. Can dogs detect any type of cancer?

Author's response

They can only detect the type of tumour that they are trained to detect. However, we only treat tumours that are difficult to diagnose with medical instruments and are usually detected in late stages. such as lung cancer and ovarian cancer. In these cases, we see this prediagnosis as a perfect complementary method.

Added to the introduction:(marked in red).  “The Center deals with tumours that are difficult to diagnose with medical instruments in the early stages (e.g. lung cancer and ovarian cancer“.

Added to the Conclusion:(marked in red).  “However, if the way of the training is modified so that the dog can work with a single sample per round, HR monitoring could become an important supplementary method. There are also other possibilities that could support this detection, such as the overall behaviour of the dog during exercise. However, further experimental planning and observations are required. The results of this study can contribute towards the improvement of the early diagnosis and, thus, the outcome of patients with lung cancer.”